# WHEN AND WHY IS PRETRAINING OBJECT-CENTRIC REPRESENTATIONS GOOD FOR REINFORCEMENT LEARNING?

## ABSTRACT

Unsupervised object-centric representation (OCR) learning has recently been drawing a lot of attention as a new paradigm of visual representation. This is because of its *potential* of being an effective pretraining technique for various downstream tasks in terms of sample efficiency, systematic generalization, and reasoning. Although image-based reinforcement learning (RL) is one of the most important and thus frequently mentioned such downstream tasks, the benefit in RL has surprisingly not been investigated systematically thus far. Instead, most of the evaluations have focused on rather indirect metrics such as segmentation quality and object property prediction accuracy. In this paper, we investigate the effectiveness of OCR pretraining for image-based reinforcement learning via empirical experiments. For systematic evaluation, we introduce a simple object-centric visual RL benchmark and verify a series of hypotheses answering questions such as "Does OCR pretraining provide better sample efficiency?", "Which types of RL tasks benefit most from OCR pretraining?", and "Can OCR pretraining help with out-of-distribution generalization?". The results suggest that OCR pretraining is particularly effective in tasks where the relationship between objects is important, improving both task performance and sample efficiency when compared to single-vector representations. Furthermore, OCR models facilitate generalization to out-of-distribution tasks such as changing the number of objects or the appearance of the objects in the scene.

## 1 INTRODUCTION

Motivated by the natural ability of humans to break down complex scenes into their constituent entities and reason about them, there has been a surge of recent research in learning unsupervised object-centric (OCR) representations (Eslami et al., 2016; Crawford & Pineau, 2019; Kosiorek et al., 2018; Lin et al., 2019; Jiang et al., 2019; Kipf et al., 2019; Veerapaneni et al., 2020; Burgess et al., 2019; Greff et al., 2019; Engelcke et al., 2019; 2021; Locatello et al., 2020; Singh et al., 2021; Kipf et al., 2021; Elsayed et al., 2022; Singh et al., 2022). These approaches learn a structured visual representation of a scene, modeling an image as a composition of objects. By using object-centric representations, downstream tasks can potentially benefit from improved systematic generalization, better sample efficiency, and the ability to do reasoning between the objects in the scene.

Since these representations can be obtained from visual inputs without the need for explicit labels, they have the promise of being an effective pretraining technique for various downstream tasks, including reinforcement learning (RL). However, most previous works in this line of research have evaluated OCRs only in the context of reconstruction loss, segmentation quality, or property prediction accuracy (Dittadi et al., 2021). While several studies have attempted to apply OCR to RL (Goyal et al., 2019; Zadaianchuk et al., 2020; Watters et al., 2019b; Carvalho et al., 2020), OCR pretraining has not been evaluated for RL tasks systematically and thoroughly. Watters et al. (2019b) evaluates OCR pretraining for a synthetic benchmark but a simple search is used rather than policy learning and less complex tasks are evaluated than our benchmark (e.g., the distractors can be ignored while our task requires the agent to avoid distractors).

In this study, we investigate *when and why OCR pretraining is good for RL*. To do this, we propose a new benchmark to cover many object-centric tasks such as object interaction or relational reasoning. Applying OCR pretraining to this benchmark, we empirically verify a series of hypotheses about decomposed representations that have been discussed previously but not systematically investigated (van Steenkiste et al., 2019; Lake et al., 2017; Greff et al., 2020; Diuk et al., 2008; Kansky et al., 2017; Zambaldi et al., 2018; Mambelli et al., 2022; Goyal et al., 2019; Carvalho et al., 2020; Zadaianchuk et al., 2020). For example, our experiments provide answers to questions such as: "Can decomposed representations improve the sample efficiency?", "Can decomposed representations help with the out-of-distribution generalization?", and "Can decomposed representations be helpful to solve relational reasoning tasks?". Furthermore, we thoroughly investigate the important characteristics of applying OCR to RL, such as how number of objects in the scene affects RL performance, which OCR models work best for RL, and what kind of pooling layer is appropriate to aggregate the object representations.

The main contribution of this paper is to provide empirical evidence about the long-standing belief that object-centric representation learning is useful for reinforcement learning. For this, we have the following more specific contributions: (1) Propose a new simple benchmark to validate OCR pretraining for RL tasks systematically, (2) Evaluate OCR pretraining performance compared with various baselines on this benchmark, and (3) Systematically analyze different aspects of OCR pretraining to develop a better understanding of when and why OCR pretraining is good for RL. Lastly, we will release the benchmark and our experiment framework code to the community.

## 2 RELATED WORK

**Object-Centric Representation Learning**. Many recent works have studied the problem of obtaining object-centric representations without supervision (Eslami et al., 2016; Crawford & Pineau, 2019; Kosiorek et al., 2018; Lin et al., 2019; Jiang et al., 2019; Kipf et al., 2019; Veerapaneni et al., 2020; Burgess et al., 2019; Greff et al., 2019; Engelcke et al., 2019; 2021; Locatello et al., 2020; Lin et al., 2020; Singh et al., 2021; Kipf et al., 2021; Elsayed et al., 2022; Singh et al., 2022). These works are motivated by the potential benefits to downstream tasks such as better generalization and relational reasoning (Greff et al., 2020; van Steenkiste et al., 2019).

There are two main methods for building slot representations; bounding box based methods (Eslami et al., 2016; Crawford & Pineau, 2019; Kosiorek et al., 2018; Lin et al., 2019; Jiang et al., 2019) or segmentation based methods (Kipf et al., 2019; Veerapaneni et al., 2020; Burgess et al., 2019; Greff et al., 2019; Engelcke et al., 2019; 2021; Locatello et al., 2020; Singh et al., 2021; Kipf et al., 2021; Elsayed et al., 2022; Singh et al., 2022). The bounding box based methods infer the latent variables for object presence, object location, and object appearance temporally (Eslami et al., 2016; Kosiorek et al., 2018) or spacially (Crawford & Pineau, 2019; Lin et al., 2019; Jiang et al., 2019). These methods work best for objects of regular shape and size. Segmentation-based methods are more flexible than bounding box based methods and have shown good performance for natural scenes or videos (Singh et al., 2021; 2022; Kipf et al., 2021; Elsayed et al., 2022). In this study, we evaluated the segmentation based models only, because of their possibility to be applied to more natural tasks.

**Object-Centric Representations and Reinforcement Learning**. RL is one of the most important and frequently mentioned downstream tasks where OCR is thought to be helpful. This is because it has been previously shown that applying decompositional representation to RL can perform better generalization and reasoning and learn more efficiently (Zambaldi et al., 2018; Garnelo et al., 2016; Diuk et al., 2008; Kansky et al., 2017; Stanić et al.; Mambelli et al., 2022; Heravi et al., 2022). However, to our knowledge, there have been no studies that systematically and thoroughly show these benefits. Goyal et al. (2019) evaluated OCR for RL by learning end-to-end. Through end-to-end learning, OCR learns a task-specific representation, which may be difficult to apply to other tasks and may not have the various strengths obtained through unsupervised OCR learning such as sample efficiency, generalization, and reasoning. Zadaianchuk et al. (2020) investigated OCR pretraining, but applies the bounding box based method (Jiang et al., 2019) and proposes/evaluates a new policy for the limited regime; goal-conditioned RL. Watters et al. (2019b) trained OCR with the exploration policy.

RL methods using decomposed representations have been previously investigated (Zambaldi et al., 2018; Garnelo et al., 2016; Diuk et al., 2008; Kansky et al., 2017; Stanić et al.; Mambelli et al., 2022; Heravi et al., 2022). Many of these works use CNN feature maps (Zambaldi et al., 2018; Stanić et al.; Heravi et al., 2022) as the representation or their own encoders (Garnelo et al., 2016). In other studies, ground truth states have been used (Diuk et al., 2008; Kansky et al., 2017; Mambelli et al., 2022), and these representations have been implemented through separate object detectors and encoders (Diuk et al., 2008; Carvalho et al., 2020). The effectiveness of the pretrained representations for out-of-distribution generalization of RL agents is studied in (Träuble et al., 2021), but only the distributed representation (e.g., VAE) is evaluated. In our work, we use a similar model as a baseline to compare with our pretrained OCR model.

## 3 EXPERIMENTAL SETUP

In this section, we provide an overview of our experimental setup. We first discuss the OCR models and baselines we chose to evaluate and explain how the representations are used in a policy for RL. We then introduce the tasks used in our experiments detailing the motivation behind each task.

### 3.1 MODELS

Each model consists of (1) an **encoder** that takes as input an image observation and outputs a latent representation and (2) a **pooling layer** that combines the latent representation into a single vector suitable to be used for the value function and policy network of an RL algorithm. We use PPO (Schulman et al., 2017) for all our experiments. Detailed information about the architecture is in Appendix E.

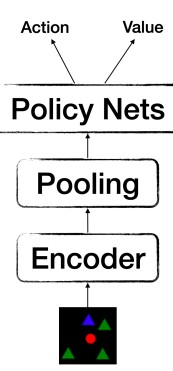

Figure 1: The general architecture of the agent

**Encoders.** We evaluated three state-of-the art OCR models, IODINE (Greff et al., 2019), Slot-Attention (Locatello et al., 2020), and SLATE (Singh et al., 2021). As non-OCR baselines, we pretrained a $\beta-$VAE Higgins et al. (2016) and evaluated both the single-vector representation (VAE) as well as the CNN feature map (CNN(VAE)). These encoders are pretrained on datasets consisting of scenes with randomly distributed objects or observations of trajectories taken by a random policy in the environment. Then, the encoders are frozen and only the pooling layers, policy, and value networks are updated when the agent is trained with RL. Lastly, we also compared these pretrained models with policies trained end-to-end from pixels (E2E CNN) and ground truth state (GT).

**Pooling Layers.** In order to use the different types of representations for RL, we implement two types of pooling layers that combine the representations into a single vector used for the value function and policy network. The MLP pooling layer combines the representations using a multi-layer perceptron. The transformer pooling layer uses a vanilla Transformer encoder (Vaswani et al., 2017). Similar to Stanić et al., the output of a CLS token is used as the input for the policy. For CNN(VAE), the cells of the CNN feature map are used as input tokens and a positional embedding is added. For the OCR and GT models, the positional embedding is not used since those latent representations are order-invariant. For the E2E CNN model, the output of the CNN is used directly in the policy so no pooling layer is used.

### 3.2 BENCHMARK AND TASKS

To verify our hypotheses, we created a suite of tasks using objects from Spriteworld (Watters et al., 2019a) (Figures 2a-d). While this environment is visually simple, we wanted to ensure that the OCR models could cleanly segment the objects of the scene into separate slots, reducing the possibility that the downstream RL performance is affected by poor OCR quality.

In order to evaluate the performance of OCR pretraining on a more visually complex environment, we also implemented a robotic reaching task using the CausalWorld framework (Ahmed et al., 2021) (Figure 2e). Details about the implementation of these benchmarks can be found in Appendix A.

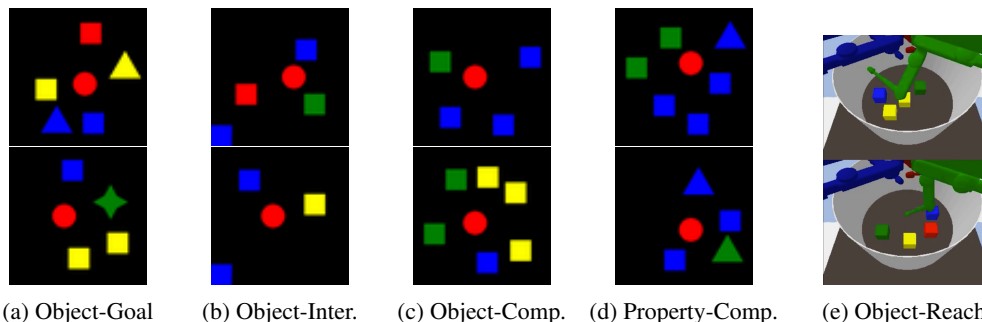

(a) Object-Goal    (b) Object-Inter.    (c) Object-Comp.    (d) Property-Comp.    (e) Object-Reach.

Figure 2: Samples from the five tasks in our benchmark; Object Goal / Object Interaction / Object Comparison / Property Comparison / Object Reaching tasks. In the 2D tasks ((a) - (d)), the red ball is always the agent. See main text for details about each task. In the robotics task (e), the goal is to use the green robotic finger to touch the blue object before touching any of the distractor objects.

**Object Goal Task:** In this task, the agent (red circle), target object (blue square), and other distractor objects are randomly placed. The goal of the task is for the agent to move to the target object without touching any of the distractor objects. Once the agent reaches the target object, a positive reward is given and the episode ends. If a distractor object is reached, the episode ends without any reward. The discrete action space consists of the four cardinal directions to move the agent. To solve this task, the agent must be able to extract information about the location of the target object as well as the objects in between the agent and the target. Therefore, through this task, we can verify that the agent is able to extract per-object information from the representation.

**Object Interaction Task:** This task is similar to the object goal task, but requires the agent to push the target to a specific location. In Figure 2b, the bottom left blue square area is the goal area. Since the agent cannot push two objects at once, the agent must plan how to move the target to the goal area while avoiding the other objects. Therefore, through this task, we can verify not only how well the agent can extract per-object information, but also how well the agent can reason about how the objects interact. The action space is the same as above, and the reward is only given when the agent pushes the target to the goal area.

**Object Comparison Task:** This task is designed to test relational reasoning ability and is motivated from the odd-one-out task in cognitive science Crutch et al. (2009); Stephens & Navarro (2008); Beatty & Vartanian (2015), which has been previously investigated with language-augmented agents Lampinen et al. (2022). To solve this task, the agent must determine which object is different from the other objects and move to it. That is, it must find the object that does not have any duplicates in the scene. Unlike the object goal or object interaction tasks, the characteristics of the target object can change from episode to episode. For example, in Figure 2c, the green box is the target object in the top sample, while the blue box is the target object in the bottom sample. Therefore, in order to know which object is the target, the agent must compare every object with every other object, which requires object-wise reasoning. The action space and reward structure are the same as the Object Goal Task.

**Property Comparison Task:** This task is similar to the Object Comparison Task, but requires more complex reasoning. The agent must find the object with a *property* (i.e. color or shape) that is different from the other objects. For example, in the top sample of Figure 2d, the blue triangle is the target because it is the only triangle in the scene. In the bottom sample, the green triangle is the target because it is the only object that is green. Therefore, this task requires property-level comparison, not just object-level comparison. While OCRs are designed to be disentangled at the object-level, it is not obvious how easily specific properties can be extracted and used for reasoning. Through this task, we can verify how well OCRs can facilitate property-level reasoning. The action space and reward structure are the same as the Object Comparison Task.

**Object Reaching Task:** Lastly, in order to evaluate the models in a more visually realistic environment, we also created a version of the Object Goal Task using the CausalWorld framework (Ahmed et al., 2021) (Figure 2e). In this environment, a fixed target object and a set of distractor objects are randomly placed in the scene. The agent controls a tri-finger robot and must reach the target object with one of its fingers (the other two fingers are always fixed) to obtain a positive reward and solve

the task. If the finger first touches one of the distractor objects, the episode ends without any reward. The action space in this environment consists of the 3 continuous joint positions of the moveable finger. We do not provide proprioceptive information to the agent, so it must learn from images how to control the finger.

## 4 EXPERIMENTS

We first outline three key hypotheses in our study, building upon previous work on relational modeling of entities for reinforcement learning (Diuk et al., 2008; Kansky et al., 2017; Zambaldi et al., 2018; Mambelli et al., 2022; Goyal et al., 2019; Carvalho et al., 2020). We present experimental evidence supporting these hypotheses and answer some interesting questions that arose during our investigations. SLATE (Singh et al., 2021) is used to represent OCR pretraining models, and a comparison with other OCR models is shown in Figure 7. For OCR models, and CNN(VAE), Transformer pooling layer is used. For GT, we used MLP pooling layer for object interaction task and Transformer pooling layer for other tasks, because GT with MLP pooling layer shows better performance for object interaction task. For end-to-end CNN (E2E CNN) and VAE, MLP pooling layer is used. The result is the averaged performance from three random seeds, and the mean or mode of the action distribution is used as the action for evaluation.

### 4.1 KEY HYPOTHESES

**Hypothesis 1 (H1):** *OCR pretraining can improve the sample efficiency of agent learning.* Since OCR pretraining provides a representation for each object, if the task is object-centric and OCR pretraining can obtain good representations, it is reasonable to believe an agent using OCR pretraining can learn more efficiently than agents using other representations (Zadaianchuk et al., 2020). We first investigate this general belief that has not yet been investigated systematically.

**Hypothesis 2 (H2):** *OCR pretraining can be beneficial to solve relational reasoning tasks.* It has been considered by several previous works that a decompositional representation will be useful for reasoning (Greff et al., 2020; van Steenkiste et al., 2019; Lake et al., 2017) and this has been experimentally verified (Zambaldi et al., 2018; Carvalho et al., 2020). Since OCR pretraining provides a well-decomposed representation per object, we can expect that OCR pretraining will be advantageous for the reasoning tasks.

**Hypothesis 3 (H3):** *OCR pretraining can help in generalization of agents.* Due to its object-wise modular representations, OCR has been shown to generalize well to out-of-distribution data such as unseen number of objects or unseen combination of objects (Dittadi et al., 2021; Locatello et al., 2020; Greff et al., 2019; Singh et al., 2021). Agents using explicit interaction networks like Transformers (Zambaldi et al., 2018) or Linear Relational Networks (Mambelli et al., 2022) have also shown good generalization performance in policy learning. It stands to reason, then, that combining OCR pretraining with a Transformer pooling layer should also have better generalization to out-of-distribution data in RL.

#### 4.1.1 SAMPLE EFFICIENCY AND RELATIONAL REASONING (H1 AND H2)

As shown in Figure 3, OCR pretraining outperforms VAE pretraining and its variant CNN (VAE). In fact, the agents using VAE pretraining are not able to solve any of these tasks, pointing to the importance of OCR for this set of tasks. When compared with the end-to-end CNN (E2E CNN), OCR pretraining shows similar sample efficiency for the Object Goal and Object Interaction tasks,

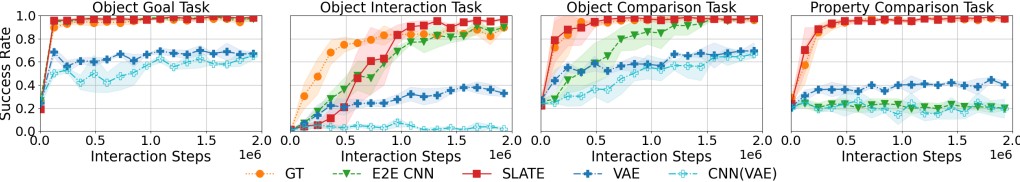

Figure 3: Success rate vs Interaction steps for the Object Goal / Object Interaction / Object Comparison / Property Prediction tasks. There are 2 objects in the Object Interaction task environment, and 4 objects in the other environments.

but learns significantly more quickly in the Object Comparison and Property Comparison tasks (E2E CNN is not able to solve the Property Comparison task within 2 million steps). It is very inefficient for the E2E CNN agent to learn the relationships between the objects only from the sparse reward, whereas the OCR pretraining agent can leverage the modular representations of the objects in the scene. This result supports our hypothesis that OCR pretraining can improve sample efficiency, and is especially helpful in the two tasks that require more relational reasoning among objects.

Interestingly, SLATE shows comparable performance with the ground truth state agent (GT) for the Property Comparison task, which requires reasoning at the property level. This is despite the fact that SLATE representations are not necessarily disentangled at the property level. We hypothesize that the transformer pooling layer plays a critical role in correctly extracting the property level information. We also find that in this case, the hyperparameters have an important effect, which we will discuss in **Q5**.

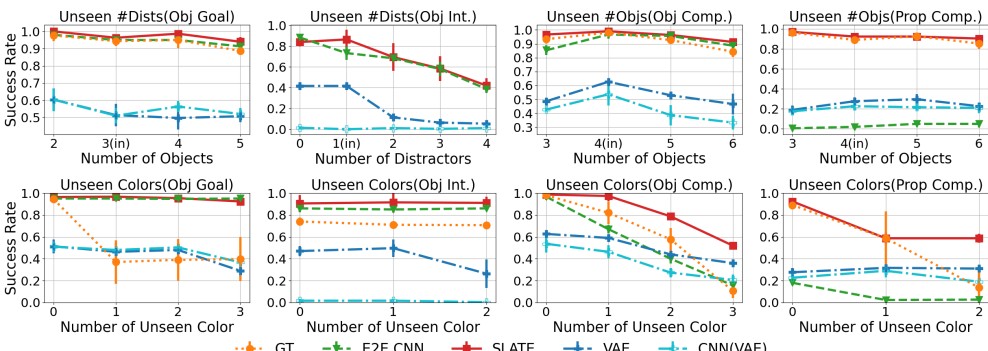

Figure 4: Generalization performance for the out-of-distribution settings. **Top row:** The success rate for the unseen number of objects. The in-distribution setting is denoted by "(in)". To clearly show the performance difference between models, we only include the models that can reasonably solve the task. The same figure with all the models is in Figure 8. **Bottom row:** The success rate for unseen object colors. The leftmost points in each chart correspond to the in-distribution task.

### 4.1.2 OUT-OF-DISTRIBUTION GENERALIZATION (H3)

In order to test the effect of OCR pretraining on generalization, we evaluated three out-of-distribution cases: unseen number of objects, unseen object colors, and unseen combinations of objects. Additional out-of-distribution results are discussed in Appendix C.1 and C.2.

First, we investigate the effect on agent performance when the number of objects differs from that on which the agent was trained. The results are shown in the top row of Figure 4. We see that for the Object Goal, Object Comparison, and Property Comparison tasks, OCR pretraining (SLATE) can generally maintain good performance as the number of objects in the scene changes. The performance on the Object Interaction task, however, degrades significantly as the number of distractor objects increases. Since the agent must learn to avoid the other objects as it pushes the target object to the goal, this task because much harder as the number of objects increases.

Interestingly, we find that the E2E CNN can also maintain good performance as the number of objects changes. This makes sense for the Object Goal task, since the CNN is able to extract more objects in the scene than it was trained on. The target object, then, which is always the same color, can also easily be extracted. For the Object Comparison task, however, increasing the number of objects can cause unseen patterns such as the ones in Figure 2c. To solve the task, each object needs to be compared with all the other objects to find the odd-one-out. The Transformer Pooling layer can handle this pairwise comparison so it makes sense that SLATE and GT perform well when scaling to more objects. The fact that the E2E CNN also scales well to more objects suggests that the multi-layered CNN can also select the odd-one-out object; however, the failure of the E2E CNN on the Property Comparison task indicates that this ability is limited as the task becomes more difficult.

Next, we investigate agent performance when evaluated on object colors not seen during training. The experimental details such as which colors are changed are described in Appendix B.1. The

results are shown in the bottom row of Figure 4. In this case, we see the pretrained OCR model (SLATE) generalizes the best when compared to the baselines, although performance does degrade as the number of unseen properties increases. Except for the Object Interaction task, GT is not robust to unseen object types since the object properties are represented as integer indices and the networks are not robust to indices it has not seen during training. For the Object Interaction task, GT generalizes better than other tasks, because MLP pooling uses concatenated GT state in the order of agent, target object, and distractor for this task. We notice that both SLATE and E2E CNN generalize almost perfectly to unseen colors in the Object Goal and Object Interaction tasks. This makes sense because only the distractor object colors change in this scenario and it is not important what colors they are. For the Object Comparison task, however, the colors of the objects are important and there is a sharp decrease in performance for both SLATE and E2E CNN as the number of unseen colors increases. Note, however, that for the Object-Comparison task, when 2 colors are changed in the color set, every scene becomes previously unseen. Yet SLATE still achieved around 80% success rate indicating the robustness of OCR pretraining.

Lastly, we investigated the effect of unseen combinations of objects, meaning all the object types are shown in training, but the combinations of object types on which we evaluate are not. This compositional ability is one of the strengths of decompositional representations (Greff et al., 2020), so we wanted to investigate it in the context of RL. We evaluate on the Object Comparison task and the results are shown in Table 1.

|  | Success Rate | |
| Model | ID | OOD |
| --- | --- | --- |
| GT | $0.94 \pm 0.008$ | $\mathbf{0.44} \pm 0.038$ |
| E2E CNN | $\mathbf{0.96} \pm 0.015$ | $0.116 \pm 0.04$ |
| SLATE | $0.949 \pm 0.011$ | $0.159 \pm 0.026$ |

Table 1: Generalization performance of the Object-Comparison Task for unseen combinations.

OCR pretraining is worse than GT, even the success rate is lower than random gussing on 4 objects. Interestingly, this is lower than when all the objects are unseen as shown in Figure 4. In that setting, when the number of unseen colors is 3, every object is previously unseen. We hypothesize that this result is because the *difference* between the slots is important to solve this task, so even though the objects are seen in training, if the difference is smaller than it has previously seen, the agent cannot solve it well. The detail settings and results are described in B.2 and Table 3.

## 4.2 ANALYSIS

The remainder of this section answers several important questions that probe different aspects of OCR pretraining for RL.

**Question 1:** *Can OCR pretraining work better than the baselines in environments with more objects? What happens if there are fewer objects?* The motivation for this question is related to the binding problem in neural networks (Greff et al., 2020). The non-OCR baselines have distributed representations, so the more objects there are in the scene, the more information needs to be bound together to represent them as entities. On the other hand, OCRs are free from this problem since they scale with the number of objects in a scene.

| Task | #Objs | Models | | Task | #Objs | Models | |
| | | SLATE | E2E CNN | | | SLATE | VAE |
| --- | --- | --- | --- | --- | --- | --- | --- |
| Object | 4 | $0.979 \pm 0.01$ | $\mathbf{0.985} \pm 0.01$ | Object | 1 | $0.997 \pm 0.01$ | $\mathbf{0.999} \pm 0.00$ |
| Goal | 6 | $\mathbf{0.95} \pm 0.00$ | $0.746 \pm 0.00$ | Goal | 3 | $\mathbf{0.992} \pm 0.01$ | $0.686 \pm 0.02$ |
| Object | 4 | $\mathbf{0.985} \pm 0.01$ | $0.97 \pm 0.01$ | Object | 1 | $\mathbf{0.99} \pm 0.01$ | $0.971 \pm 0.01$ |
| Comp. | 6 | $\mathbf{0.823} \pm 0.00$ | $0.34 \pm 0.06$ | Int. | 3 | $\mathbf{0.859} \pm 0.03$ | $0.345 \pm 0.08$ |

(a) More number of objects

(b) Fewer number of objects

Table 2: Performance comparison when more or fewer objects are in the environments.

To answer this question empirically, we first evaluated SLATE and E2E CNN for the Object Goal and Object Comparison tasks by increasing the number of objects. Note that VAE is not compared, since it already does not perform well with only 4 objects. The result is shown in Table 2a. As the

number of objects increases, both models have lower success rates, but the performance degradation of E2E CNN is much greater than that of SLATE. When the number of objects is 6, E2E CNN only has a success rate of around 34%. Interestingly, when learning in an environment with 4 objects and testing in an environment with 6 objects, E2E CNN showed a higher success rate of close to 90% in Figure 4 (first row, third column). One explanation for this is that the E2E CNN learns the rules from the easier tasks and can work to some extent when applied to the more difficult task, but it is not able to learn from the difficult task itself. On the other hand, under the same conditions, SLATE showed a success rate of about 82% without any help from training on an easier task.

What happens if there are fewer objects in the environment? In table 2b, we see that with fewer objects in the environment, both VAE and SLATE performed better, and the difference was much greater with VAE. Interestingly, when there is only one object, VAE showed similar performance to SLATE in both Object Goal and Object Interaction tasks. This, together with the above results, clearly shows the binding problem of the distributed representation (Greff et al., 2020).

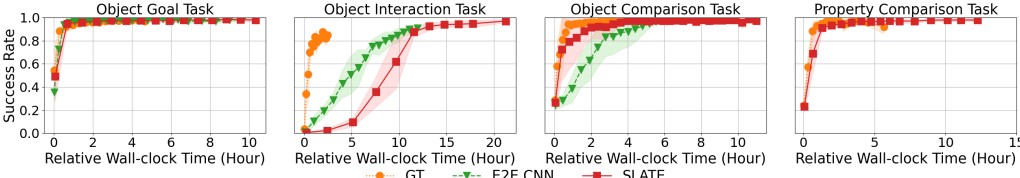

Figure 5: Success rate comparison vs wall-clock time. Each method is trained to 2 million interaction steps. VAE and CNN (VAE) were not compared because they did not show comparable performance.

**Question 2:** *Is OCR pretraining more efficient than end-to-end training in terms of wall-clock time?* While we previously discussed the better sample efficiency of OCR, since OCR models typically require more computation than CNN architectures, it is natural to ask this question. As shown in Figure 5, SLATE is slower than E2E CNN or GT in terms of time to reach 2 million steps. In terms of wall clock time to convergence, however, SLATE is comparable to GT except for the Object Interaction task. For this task, the MLP pooling layer is used for GT, which is much faster than the Transformer pooling layer. When comparing with E2E CNN, for the Object Comparison task, the gap between SLATE is lower than when comparing via interaction steps, although SLATE still does train more quickly than E2E CNN. However, for the Object Interaction task, E2E CNN trains more quickly in terms of wall-clock time, even though it takes more interaction steps.

**Question 3:** *Does OCR pretraining work well in visually complex environments where segmentation is difficult?* To answer this question, we tested SLATE and baselines in the Object Reaching task. As shown in Figure 13, the SLATE segmentation is not perfect, sometimes splitting several objects between slots and not cleanly capturing the robotic finger.

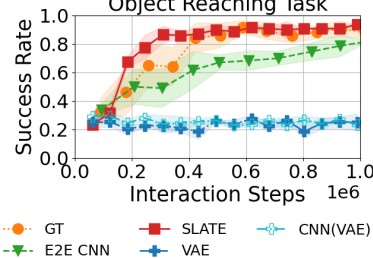

Figure 6 shows the results of this experiment. Although the segmentation is not good, the agent using the SLATE OCR performs best on this task both in terms of sample efficiency and converged success rate. Note that even though this task does not explicitly require reasoning among the objects, it is still important for the agent to learn to avoid touching the dis-

Figure 6: Success rates for the Object Reaching Task.

tractor objects before the target object, so the distance between the objects as well as the robotic finger and the objects is important to solve the task. This result suggests that the conclusions from the experiments on visually simple environments can potentially generalize to more complex environments.

**Question 4:** *Which OCR model is better for RL?* To answer this question, we evaluated three state-of-the-art OCR models, Slot-Attention (Locatello et al., 2020), IODINE (Greff et al., 2019) and SLATE (Singh et al., 2021). For SLATE, we used the version using CNN encoder mentioned as an ablation of that in Singh et al. (2021). The detailed configurations are in Appendix E.

Figure 7 shows that SLATE outperforms Slot-Attention and IODINE in these tasks. Even Slot-Attention and IODINE fail to solve the Object Interaction task. Interestingly, SLATE and Slot-

Figure 7: Success rates comparison between different OCR models.

Attention are very similar architecturally—the main difference is in the decoder and slot size, which is larger for SLATE. To investigate further, we also test Slot-Attention with a larger slot size to match SLATE (Slot-Attention (Large)).

Slot-Attention (Large) shows similar performance to SLATE, and outperforms Slot-Attention in all tasks. When looking at segmentation quality (Table 5) and property prediction accuracy (Figure 11), Slot-Attention and Slot-Attention (Large) do not show significant differences. From this result, at least for Slot-Attention, slot size appears to be important for RL performance regardless of segmentation quality or property prediction accuracy. For those models, we evaluated the correlation of RL performance with segmentation quality, property prediction accuracy, reconstruction loss in Appendix C.3.

**Question 5:** *How does the choice of pooling layer affect task performance?* We have generally used the Transformer pooling layer for our OCR models since (a) it is permutation invariant, which is an important property since the slots in OCR generally do not have strict ordering, and (b) it explicitly models interactions between the slot which is important for the relational reasoning tasks. As an ablation study, we applied the MLP pooling layer by concatenating all the slots and compared it with the Transformer pooling layer. As shown in Figure 7, the version that applied the MLP pooling layer to SLATE, denoted as SLATE-MLP, performed worse than when the Transformer pooling layer was used for all tasks, failing to solve the interaction and comparison tasks completely. Surprisingly, SLATE-MLP is still able to achieve good performance on the Object Goal task, despite the fact that MLPs are not permutation invariant. This may be because the task is easier than others, and the target object can still be extracted from the MLP and the interaction between objects is not very important to solve that task.

## 5 CONCLUSION AND DISCUSSION

In this paper, we investigated *when and why OCR pretraining is good for RL*. To do this, we designed a new benchmark and empirically evaluated several OCR models and baselines through various conditions. Through this study, we found several conditions that OCR pretraining is good for RL:

- OCR pretraining performs better than distributed representations when the relationships between objects are important to solving the task. However, it can be slower for tasks where object-wise reasoning is not required due to computational overhead.

- Because OCR pretraining provides a separate representation for each object between slots, it performs better than distributed representations in an environment with many objects.

- OCR is better for generalization of agents, especially for objects not seen during training, showing better performance than end-to-end learned representations.

- OCR pretraining showed better performance than baselines even in visually complex environments where segmentation is not perfect. This suggests that what we found in visually simple environments can be applied to more complex environments.

Although our benchmark covers several important object-centric tasks, such as object interaction and relational reasoning, there are other aspects of agent learning that can benefit from OCR such as partially observable environments or tasks that require exploration. These are good candidates to extend the benchmark in the future. Additionally, while we used several strong OCR baselines in our experiments, there are other OCR models that may be investigated in the future.

We hope that our benchmark can be useful for evaluating OCR models in the context of agent learning, in addition to the previously standard metrics such as segmentation quality and property prediction accuracy. Further discussion is in Appendix D.

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

# A BENCHMARK

Our benchmark consists of 2D tasks from the Spriteworld (Watters et al., 2019a) and a 3D task from the CausalWorld (Ahmed et al., 2021).

## A.1 2D TASKS

On 2D tasks, the observation size and channels are 64 and 3. The object size is represented as the percentage of the observation size. There is no occlusion between objects, and the agent is always the red ball with a size of 0.15. At the beginning of the episode, the agent position is always in the center. The action set is to move up/down/left/right. At each action, the agent moves 0.05 to the direction. The objects are randomly distributed, and their characteristics are sampled from the given set by following the rules of each task. For tasks, The object sizes are 0.15.

For pretraining, non-task sepcific dataset is used. The dataset for pretraining is consisted of scenes with randomly distributed objects. The number of objects in the scene is 5 and the object color is one of [blue, green, yellow, red] and shape is one of [square, triangle, star_4, circle]. The size is one of [0.15, 0.22]. The minimum distance between object is 0.15, so when object size is 0.22, occlusion could happen. The numbers of scenes for training and validation are 1 million and $100,000$.

### A.1.1 OBJECT GOAL TASK

The sets for shape, color are [square, triangle, star_4], and [blue, green, yellow, red]. The target object is always blue square. At every episode, only one target object is in the environment, and other objects are randomly generated.

### A.1.2 OBJECT INTERACTION TASK

The color set is [blue, green, yellow] and the shape is fixed as square. The target object is always blue square. To make a solvable task always, every object is far from the wall at least the object size (then the agent can push the target in any direction).

### A.1.3 OBJECT COMPARISON TASK

The color set is [blue, green, yellow], and the shape is fixed as a square. There must be a single unique object, and other objects are randomly generated in this rule.

### A.1.4 PROPERTY COMPARISON TASK

The shape and color sets are [square, triangle] and [blue, green]. There must be only one unique property (e.g., only one square in the environment or only one blue object), and other objects are randomly generated in this rule.

## A.2 3D OBJECT REACHING TASK

The tri-finger environment is modified so that two of the fingers are always in the upright positions. The actions only affect the third green finger. At the beginning of each episode, all the fingers are in the upright positions. The observation size and channels are 64 and 3.

The color set [blue, green, yellow, red] and all objects are cubes.

The target object is always blue. For each episode, there is only one target object and the other objects are randomly generated. The positions of the objects are randomly chosen and chosen so that the blocks do not overlap with each other.

For pretraining, the dataset collected through random policy is used. The number of observations to train is 1 million.

Figure 8: The unseen number of objects test results including VAE and CNN (VAE). There are no GT tests for object interaction tasks. This is because the MLP pooling layer used for GT cannot be applied to the unseen number of objects.

| Models | [B,G],[G,Y]→[B,Y] | | [B,G],[B,Y]→[G,Y] | | [B,Y],[G,Y]→[B,G] | |
|---|---|---|---|---|---|---|
| | ID | OOD | ID | OOD | ID | OOD |
| GT | $0.95 \pm 0.1$ | $\mathbf{0.867} \pm 0.05$ | $0.96 \pm 0.01$ | $\mathbf{0.23} \pm 0.035$ | $0.927 \pm 0.006$ | $\mathbf{0.22} \pm 0.03$ |
| E2E CNN | $\mathbf{0.957} \pm 0.06$ | $0.117 \pm 0.015$ | $\mathbf{0.97} \pm 0.017$ | $0.123 \pm 0.067$ | $\mathbf{0.953} \pm 0.012$ | $0.107 \pm 0.035$ |
| SLATE | $0.947 \pm 0.015$ | $0.177 \pm 0.025$ | $0.96 \pm 0.02$ | $0.147 \pm 0.021$ | $0.94 \pm 0.0$ | $0.153 \pm 0.031$ |

Table 3: The detail results of the unseen combination evaluation

## B  EXPERIMENT DETAILS

### B.1  THE UNSEEN COLOR EVALUATION

To validate the generalization of the agent with the unseen object, we use the unseen color. For Object Goal and Interaction tasks, we change the colors of the distractors only, because the target object of the tasks is always the blue square. For Object Goal task, the in-distribution color set is [blue, green, yellow, red], and it is changed one by one as the follow; [blue, green, yellow, pink] → [blue, green, brown, pink] → [blue, cyan, brown, pink]. For Object Interaction task, the in-distribution color set is [blue, green, yellow] and it is changed as [blue, green, pink] → [blue, cyan, pink].

For Object Comparison and Property Comparison tasks, we change the entire object color. For Object Comparison task, the in-distribution color set is [blue, green, yellow], which is changed as [blue, green, pink] → [blue, cyan, pink] → [brown, cyan, pink]. For Property Comparison task, the in-distribution color set is [blue, green]. It is changed as [blue, pink] → [cyan, pink].

### B.2  THE UNSEEN COMBINATION EVALUATION

For this evaluation, we used Object Comparison task. The task is to select the odd-one from 4 objects and the color set is [blue, green, yellow]. To evaluate the performance on the unseen combination, we train the models for 3 tasks where [blue, green] or [blue, yellow], or [green, yellow] combinations are not given. After training the models on the tasks, we evaluate the models on the unseen combinations. The detail results are in Table 3.

## C  ADDITIONAL RESULTS

### C.1  EVALUATION WHEN THE ENVIRONMENT IS IN-DISTRIBUTION FOR OCR/VAE PRETRAINING BUT OUT-OF-DISTRIBUTION FOR THE AGENT

In the paper, we discussed the generalization of the agents when the environment is out-of-distribution for pretraining and agents. Then, what happens if the agent is tested for the environment that is in-distribution for OCR or VAE (Dittadi et al., 2021)? We evaluate it by training OCR and VAE with a dataset consisting a super set of the task (e.g., if blue and green were used in the task, then training with blue, green, yellow and pink). After that, we trained the agent on property comparison task, and tested it for the unseen shapes or colors for the agent, but seen for OCR or VAE in pretraining.

The results are in Table 4. The expected result would be that the agent performs better in an environment that is in-distribution to pretraining and out-of-distribution to the agent than in an environment

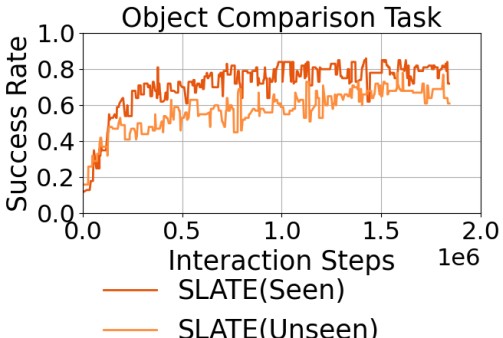

Figure 9: The performance comparison when the agents are trained on the two environments, one of which is in-distribution for OCR and another is not.

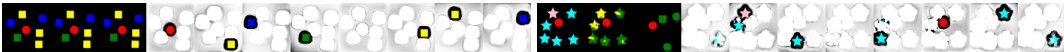

Figure 10: The **left** sample is from in-distribution and the **right** is from out-of-distribtion. Each column per sample means Observation/Reconstruction through DVAE/Reconstruction through Transformer decoder/Masks of slots (8 slots).

that is out-of-distribution to both. VAE pretraining shows better performance for the case, ID for pretraining but OOD for agent. However, interestingly, for OCR pretraining, there were no consistent results. For unseen shapes, the result is same to our expectation, but for unseen colors, the result is different from our thought. It is because even though the shape or color is seen in pretraining, the combination is new, so it could be difficult to the agent like the result for the unseen combination in Table 1.

## C.2 Agent Training when the environment is out-of-distribution for OCR pretraining

As other case, we can imagine, what happens if the agent is trained on the environment where is out-of-distribution for OCR pretraining. To evaluate this, we used OCR pretraining with 6 objects environment for object comparison task. Then, we trained the agents on the two environments, one of which is in-distribution for OCR and other is not. For out-of-distribution case, the every object in the environment is unseen, but the number of object is same to the trained dataset.

As shown in Figure 11, the performance gap is not too much, even though every object is unseen. However, interestingly, when seeing samples from each agents in Figure 10, we can find OCR not just cannot segment perfectly but also reconstruct poorly. From this result, we can find that OCR encoder can represent the objects even though the objects are unseen, which can be useful for the agent learning. It is quite related with the unseen object type results in Figure 4. The result in Figure 4 shows that the trained agent with OCR pretraining can be generalized better than baselines for the unseen object types. Those results support that we can use the OCR pretraining over the trained distribution for the agent training or the generalization of the agent.

## C.3 Do the standard metrics of evaluating OCR correlate with RL performance?

Many OCR models are evaluated through segmentation quality, reconstruction loss, or property prediction accuracy(Dittadi et al., 2021; Greff et al., 2019; Locatello et al., 2020; Singh et al., 2021; Burgess et al., 2019). Does performing well on these metrics translate to RL performance?

To answer this question, we calculated the correlations between the segmentation quality / reconstruction loss / property prediction accuracy and RL performance from SLATE / Slot-Attention / Slot-Attention (Large) / IODINE. For measuring segmentation quality and reconstruction loss, we used foreground Adjusted Rand Index (ARI) (Hubert & Arabie, 1985) and MSE. ARI, MSE, and property prediction accuracy are obtained through a dataset collected from a random policy, which

| Setting | Model | Success Rate | |
| --- | --- | --- | --- |
| | | ID for Pretraining, OOD for Agent | OOD for Pretraining and Agent |
| unseen shapes | SLATE | **0.663** ± 0.049 | **0.487** ± 0.16 |
| | VAE | 0.297 ± 0.035 | 0.297 ± 0.379 |
| unseen colors | SLATE | **0.567** ± 0.021 | **0.587** ± 0.04 |
| | VAE | 0.357 ± 0.032 | 0.31 ± 0.01 |

Table 4: The generalization performance when the setting is in-distribution for pretraining and out-of-distribution for agent learning.

| Tasks | FG-ARI | | | |
| --- | --- | --- | --- | --- |
| | SLATE | Slot-Attention | Slot-Attention (Large) | IODINE |
| Object Goal | 0.909 | 0.928 | 0.927 | 0.918 |
| Object-Interaction | 0.932 | 0.954 | 0.95 | 0.943 |
| Object Comparison | 0.912 | 0.931 | 0.93 | 0.922 |
| Property Comparison | 0.911 | 0.929 | 0.929 | 0.92 |

Table 5: Foreground ARI scores.

are shown in Tables 5, and 6, and Figure 11. For RL performance, the average of the success rate in 1000 episodes is used. The correlations are shown in Figure 12. MSE only is shown to have a positive correlation with RL performance for every task, meaning better reconstruction correlates with better RL performance. Other metrics such as ARI or property prediction accuracy are not shown to have correlated through those 4 tasks. We note that the number of validated model types is small, so it must be investigated more in future works.

| Tasks | MSE | | | |
| --- | --- | --- | --- | --- |
| | SLATE | Slot-Attention | Slot-Attention (Large) | IODINE |
| Object Goal | 13.325 | 6.5 | 6.911 | 9.345 |
| Object-Interaction | 57.412 | 35.915 | 45.961 | 85.714 |
| Object Comparison | 14.821 | 5.898 | 7.53 | 9.026 |
| Property Comparison | 10.583 | 5.539 | 6.708 | 8.5 |

Table 6: MSE for Object Goal / Object Comparison / Property Comparison tasks

## C.4 COMPARISON WITH CNN FEATURE MAPS FROM OTHER MODELS

**CNN feature map from SLATE** (Heravi et al., 2022): As another pretrained CNN feature map, we used the frozen CNN feature map from the pretrained SLATE encoder. Frozen CNN feature map from pretrained OCR is used in (Heravi et al., 2022). In the paper, the CNN feature map is encoded through another CNN which is trained through reward signal. We follows the architecture, but we evaluated also when applying Transformer on top of another CNN for pooling.

**Multiple E2E CNNs** (Kipf et al., 2019; Watters et al., 2017): As previous works (Kipf et al., 2019; Watters et al., 2017) did, we evaluated the multiple CNN encoders that are trained through the reward signal. We used 5 encoders, and Transformer pooling is applied on top of the distributed representations from each encoder.

**CNN feature map from E2E CNN (CNN(E2E CNN))**: In end-to-end learning manner, using CNN feature map not the distributed representation from the MLP on top of CNN. Similar architecture is used for relational reasoning in (Santoro et al., 2017). For pooling, Transformer is used.

**CNN feature map from SLATE (CNN(SLATE))**: Using CNN feature map from SLATE encoder. It is frozen like OCR pretraining. This CNN feature map size is too large, we used additional CNN

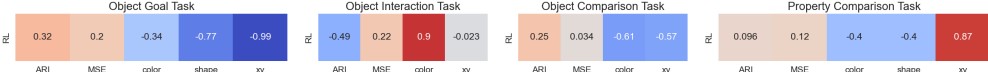

Figure 11: The property prediction accuracy for four OCR models and VAE. For color and shape, the score is accuracy (higher is better). For x and y coordinates, it is a distance between prediction and grondtruth, so lower is better.

Figure 12: The correlation between saturated / intermediate RL performance and usually used measurements for OCR (e.g., ARI, reconstruction loss or property prediction accuracy)

on top of that to collect smaller size of CNN feature map. On top of the smaller CNN feature map, Transformer pooling is used. Similar architecture is used in (Heravi et al., 2022), but they didn't use Transformer, but use MLP on top of additional CNN.

**Multiple E2E CNNs (E2E MultipleCNNs)**: The model design to use multiple CNN encoders has been used to get object representation earlier in (Kipf et al., 2019; Watters et al., 2017). In the similar way, we evaluate this architecture on our benchmark. We note that the CNNs are trained through reward. We used 5 CNNs, and Transformer pooling is used.

As shown in Table 7, CNN feature maps with Transformer pooling fail to solve any tasks except for the Object Goal task. This should be investigated more in future work, but one of our hypotheses is that it can be difficult to train the Transformer pooling with a large number of patch representations through the reward signal. One piece of evidence for this is that the model using MLP pooling on the CNN feature map from SLATE works better than the model with Transformer pooling on the same feature map for the Object Interaction task.

Multiple E2E CNNs outperform E2E CNN except for the Object Interaction task. We hypothesize that this is because the encoder model size is much larger than E2E CNN and the reward from the Object Interaction task is much sparser than other tasks. Another interesting result is that Multiple E2E CNNs solve the Property Comparison task somewhat. From this, we can expect that through multiple encoders, the model can represent object-wise information as shown in (Kipf et al., 2019; Watters et al., 2017), but it is not perfect.

## D  FURTHER DISCUSSION

To evaluate OCR pretraining in this paper, the benchmark is limited in the scope where a random policy is sufficient to collect enough diverse observations, but there are lots of cases where trajectories from a random policy is not enough to pretrain the encoder. It could be an interesting topic to train OCR with auxiliary loss while learning a policy as shown in (Hafner et al., 2019; Ha & Schmidhuber, 2018). As another future research direction of the OCR community, it is also interesting to investigate non object-centric tasks with OCR or Modulating between System 1 and System 2 modes (Kahneman, 2011).

## E  MODEL DETAILS

In this section, the architectural details about used models are introduced.

### E.1  ENCODER

As the encoder, we used several OCR models (Singh et al., 2021; Locatello et al., 2020; Greff et al., 2019), $\beta-$VAE (Higgins et al., 2016) and CNN trained through policy loss.

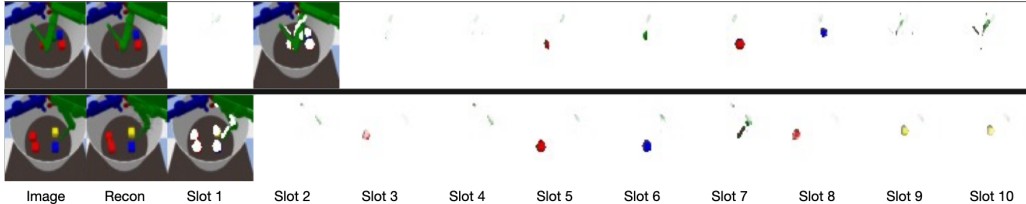

Figure 13: SLATE segmentation on the Object Reaching Task

| Tasks | Success Rate | | | |
| --- | --- | --- | --- | --- |
| | Obj. Goal | Obj. Int. | Obj. Comp. | Prop. Comp. |
| E2E CNN | $0.983 \pm 0.015$ | $0.899 \pm 0.033$ | $0.972 \pm 0.007$ | $0.208 \pm 0.037$ |
| CNN Feat from E2E CNN | $0.973 \pm 0.012$ | $0.307 \pm 0.415$ | $0.22 \pm 0.03$ | $0.185 \pm 0.038$ |
| Multiple E2E CNNs | $0.987 \pm 0.005$ | $0.222 \pm 0.209$ | $0.975 \pm 0.016$ | $0.599 \pm 0.053$ |
| SLATE | $0.977 \pm 0.006$ | $\mathbf{0.963} \pm 0.021$ | $\mathbf{0.982} \pm 0.011$ | $\mathbf{0.978} \pm 0.013$ |
| CNN Feat from SLATE with Trans pooling | $0.972 \pm 0.011$ | $0.01 \pm 0.0157$ | $0.219 \pm 0.037$ | $0.218 \pm 0.025$ |
| CNN Feat from SLATE with MLP pooling | $\mathbf{0.987} \pm 0.008$ | $0.647 \pm 0.256$ | $0.216 \pm 0.029$ | $0.224 \pm 0.028$ |

Table 7: The average success rate of 3 models with different random seeds after training to 2 million interaction steps

### E.1.1   GT

**2D tasks:** Ground Truth state of this task is given as the matrix of the number of objects $\times$ 5. Each object is represented as [COLOR index, SHAPE index, SIZE index, x coordinate, y coordinate]. The color, shape, and size indices are from a pre-specified set. The pre-specified color set is [blue, green, yellow, red, cyan, pink, brown], and the pre-specified shape set is [square, triangle, star_4, circle, pentagon, hexagon, octagon, star_5, star_6, spoke_4, spoke_5, spoke_6]. For the size, [0.15, 0.22] is given. When using the MLP pooling layer, we add some MLPs on top of GT per object state like in Table 8.

**3D Object Reaching Task:** The ground truth state of this task consists of 37 dimensions for the robot state (9 dimensions for joint positions, 9 dimensions for joint velocities, and 9 dimensions for end effector positions), and 9 dimensions for each object (3 for cartesian position, 3 for size, and 3 for color (RGB)). To represent the ground truth state of both robot state and objects in terms of slots, these 36 dimensions are concatenated together with an additional dimension indicating whether the slot is the robot arm or an object. The final representation consists of 5 slots, each with 37 dimensions.

### E.1.2   OCRs

The SLATE uses the CNN encoder originally used in (Locatello et al., 2020). Two versions of Slot-Attention are evaluated, Slot-Attention and Slot-Attention (Large) which architecture is bigger than Slot-Attention. The number of slots is different for different tasks. The hyperparamters are in Table 9 and 10.

### E.1.3   VAE

The encoder and decoder to train VAE mainly consist of multiple blocks of CNN described in Table 11 and 12. The encoder consists of 4 encoder CNN blocks and one CNN layer which channels, kernel size, stride and padding are 64, 1, 1 and 0 without activation function, respectively. The output size is $4 \times 4$, and CNN-VAE uses it as the representation of the observation. To sampling the latent variable, the CNN feature map is flatten and encoded through the linear layers of mean and variance, input and output sizes of which are 64*4*4 and 256.

To decode, the latent variable is encoded through the linear layer, which input and output sizes are 256 and 64*4*4. The decoder consists of 4 decoder blocks with pixel shuffle function between

| Type | Output Size | Activation |
|---|---|---|
| Linear | 32 | ReLU |
| Linear | 32 | ReLU |

Table 8: Hyperparameters for MLP of 2D GT state when using MLP pooling layer

| | Configurations | SLATE |
|---|---|---|
| Learning | Temp. Cooldown | 1.0 to 0.1 |
| | Temp. Cooldown Steps | 30000 |
| | LR for DVAE | 0.0003 |
| | LR for CNN Encoder | 0.0001 |
| | LR for Transformer Decoder | 0.0003 |
| | LR Warm Up Steps | 30000 |
| | LR Half Time | 250000 |
| | Dropout | 0.1 |
| | Clip | 0.05 |
| | Batch Size | 24 |
| DVAE | vocabulary size | 4 |
| CNN Encoder | Hidden Size | 64 |
| Slot Attention | Slots | - |
| | Iterations | 3 |
| | Slot Heads | 1 |
| | Slot Dim. | 192 |
| | MLP Hidden Dim. | 192 |
| | Pos Channels | 4 |
| Transformer Decoder | Layers | 4 |
| | Heads | 4 |
| | Hidden Dim. | 192 |

Table 9: Hyperparameters for SLATE

blocks, and one CNN layer which channels, kernel size, stride and padding are observation channel size, 1, 1 and 0 without activation function, respectively.

The learning rate is 0.0001, and the weight for KL-term is 5, and the batch size is 128.

### E.1.4 CNN

The CNN architecture for E2E CNN model is following the architecture used in (Mnih et al., 2015) which is also described in Table 13. After encoding the observation through this CNN, the output is flatten and pass a single linear layer with ReLU activation, which output size is 512.

### E.2 POOLING

In this study, we used two pooling layers; Transformer (Vaswani et al., 2017) and MLP which hyperparameters are described in Table 14 and 15.

### E.3 POLICY

We used PPO (Schulman et al., 2017) as the policy algorithm with the configuration in Table 16. The policy is trained through the Stable Baselines3 (Raffin et al., 2019), and the trajectories are collected through 4 environments.

| | Configurations | Slot-Attention | Slot-Attention (Large) | |
|---|---|---|---|---|
| | LR | 0.0001 | 0.0001 | |
| | LR Warm Up Steps | 30000 | 30000 | |
| Learning | LR Half Time | 250000 | 250000 | |
| | Clip | 0.05 | 0.5 | |
| | Batch Size | 24 | 24 | |
| CNN Encoder | Hidden Size | 64 | 64 | |
| | Slots | - | - | - |
| | Iterations | 7 | 3 | |
| | Slot Heads | 1 | 1 | |
| Slot Attention | Slot Dim. | 64 | 192 | |
| | MLP Hidden Dim. | 128 | 192 | |
| | Pos Channels | 4 | 4 | |

Table 10: Hyperparameters for Slot-Attention and Slot-Attention (Large)

| Channels | Kernel Size | Stride | Padding | Activation |
|---|---|---|---|---|
| 64 | 2 | 2 | 0 | ReLU |
| 64 | 1 | 1 | 0 | ReLU |
| 64 | 1 | 1 | 0 | ReLU |
| 64 | 1 | 1 | 0 | ReLU |

Table 11: Hyperparameters for VAE Encoder CNN Block

| Channels | Kernel Size | Stride | Padding | Activation |
|---|---|---|---|---|
| 64 | 3 | 2 | 1 | ReLU |
| 64 | 1 | 1 | 0 | ReLU |
| 64 | 1 | 1 | 0 | ReLU |
| 64 *4 | 1 | 1 | 1 | ReLU |

Table 12: Hyperparameters for VAE Decoder CNN Block

| Channels | Kernel Size | Stride | Padding | Activation |
|---|---|---|---|---|
| 32 | 8 | 4 | 0 | ReLU |
| 64 | 4 | 2 | 0 | ReLU |
| 64 | 3 | 1 | 0 | ReLU |

Table 13: Hyperparameters for E2E CNN

| Type | Output Size | Activation |
|---|---|---|
| Linear | 128 | ReLU |
| Linear | 128 | ReLU |

Table 14: Hyperparameters for MLP Pooling Layer

| Configurations | |
|---|---|
| Model Dim | 128 |
| heads | 8 |
| layers | 3 for GT |
| | 1 for others |

Table 15: Hyperparameters for Transformer Pooling Layer

| Configurations | |
| --- | --- |
| Steps per training | 2048 |
| LR | 0.0003 |
| coefficient for entropy term | 0.0 |

Table 16: Hyperparameters for PPO

