# OpenReview forum: "When and Why Is Pretraining Object-Centric Representations Good for Reinforcement Learning?"
_ICLR.cc/2023/Conference — Submitted to ICLR 2023_

### Official Review · Reviewer_EbVV · 2022-10-23

**Confidence:** 4
**Correctness:** 3
**Technical Novelty And Significance:** 2
**Empirical Novelty And Significance:** 2
**Recommendation:** 3

**Clarity, Quality, Novelty And Reproducibility:**

- Clarity: good, the paper is clear and easy to follow.
- Quality: ok, the presented experiments look sound but I wished the authors dove deeper into the properties of OCR.
- Novelty: incremental, I wish the authors provided more profound insights than “object-centric representations work better for object-centric tasks”.
- Reproducibility: ok, I haven’t look at the Supp. Mat. but believe the experiments are reproducible.

**Strength And Weaknesses:**

- Strengths:
    - This paper studies an important aspect of transfer in RL: what kind of representations are needed, and when? The empirical evidence makes a strong case for OCR, as they perform best on virtually all experimental testbeds (caveat: those are chosen to be object-centric, see below). I especially enjoyed the evidence from Figure 3 that OCR (ie: SLATE) can ease reasoning on relational tasks.
    - The authors are very clear in their approach (driven by hypotheses and questions), and the experiments look carefully carried out. They also thoroughly describe their results, which makes me trust their conclusions on their experiments. Altogether, I found this paper easy to follow as it is always clear what aspect of OCR is being tested.
- Weaknesses:
    - The main limitation of this work is the scope hypotheses and questions: they all take the form “does method A work better than method B in task C?”. From the title and the introduction, I was hoping to gather new insights on the “how” and the “why” OCR could be useful in RL. But no hypothesis or question ever gets to these types of insights.

        In fact, I do not think the reader ever gets an answer to the title question; the first 2 bullet points in the conclusions suggests answers, but I don’t see how those answers are supported by the analysis. For example, which experiments support that:

        - “OCR can be slower for tasks where object-wise reasoning is not important”? It looks like all testbeds require some form of object-wise reasoning.
        - “OCR provides disentangled representations for each object”? First, this property is never tested. Second, it is never compellingly shown that it is the reason for the good performance of OCR.

        For those reasons, the main message I got from this paper strikes me as somewhat incremental: “object-centric representations work better for object-centric tasks”.

    - I also have some concerns over the experimental designs.
        - First, and as mentioned above, what about non object-centric tasks? How far do OCR lag behind non-OCR pretraining? This is important as it starts to provide some ground to answer the “when does OCR work” question.
        - Second, are the E2E and VAE baselines trained on enough data and with enough variations? It would be help the paper if the authors could show that even with more data, VAE and E2E have already “plateaued” and so the benefits of OCR can’t be make up for. Similarly, what if E2E and VAE were trained on much larger datasets (eg: ImageNet) where learning OCR becomes much more challenging (because of the lack of object-centric labels).
        - Third, and a more minor point, the benchmarking testbeds look a little bit simplistic (see Fig. 2). Do the insights in the paper carry to more realistic tasks such as real-world embodied AI (Habitat/AI2Thor)? The most realistic task in this paper (object-reach) is only ever used once (Fig. 6).
    - Some minor points:
        - The background section could include more details for at least one of the OCR method (eg: SLATE) to refresh the reader’s memory. It could also outline the main differences between all OCR methods.
        - Figures are unreadable for color-blind readers or when printed black-and-white.

**Summary Of The Paper:**

- This paper investigates the qualities of object-centric representations (OCR) in the context of reinforcement learning (RL) tasks. Does so by empirically benchmarking OCR and non-OCR methods (eg: end-to-end learning (E2E) and variational auto-encoding (VAE)) on 5 tasks from 2 domains (see Fig. 2). Each benchmark is either driven by an hypothesis or a question, which are clearly stated up-front.
- The results from these empirical studies can be summarized as follows:
    - Object-centric pretraining improves sample efficiency over VAE or E2E pretraining on object-centric tasks (Section 4.1.1). But it is slower in terms of wall-clock time (Section 4.2-Q2).
    - Object-centric pretraining can learn relational properties of objects (eg: SLATE), which can enable reasoning (Section 4.1.1).
    - OCR and E2E generalize to new objects in the observation but both fail when it comes to compositionality (Sections 4.1.2, 4.2-Q1). They also generalize better to environment that are more visually complex than the ones seen during training (Section 4.2-Q3).
    - SLATE tends to outperform other OCR learning algorithms (Section 4.2-Q4) and MLP pooling degrades performance of SLATE (Section 4.2-Q5).

**Summary Of The Review:**

- Strengths:
    - The core question of the paper is important to the ICLR community.
    - The experiments are clearly laid out, and their results discussed in details.
- Weaknesses:
    - The hypotheses tested in the paper are somewhat superficial and do not answer the core question of the paper — why and when are OCR required for RL?
    - Some experimental design choices can be improved (choice of testbeds, training of baselines)?
    - (Minor) Background could be more fleshed out and figures could be more legible.

---

> ### Author Response · Authors · 2022-11-19
> **Response to Reviewer EbVV (1/3)**
>
> First of all, thank you very much for your critical and constructive feedback. We will share our response to your questions and discuss your concerns.
>
> > The main limitation of this work is the scope hypotheses and questions: they all take the form “does method A work better than method B in task C?”. From the title and the introduction, I was hoping to gather new insights on the “how” and the “why” OCR could be useful in RL. But no hypothesis or question ever gets to these types of insights.
> >
>
> Thank you for sharing your concerns. Yes, we took the form to compare different types of models, but from those comparisons, we gain insight into “when”, “why”, and “how” OCR could be useful. For example, in Hypothesis 2, we found OCR is good for reasoning tasks (”when”). To probe at the reason (”why”), we analyze the results by comparing it with different types of models (eg. E2E CNN). Through Question 5, we discussed the effect of the choice of pooling, and why utilizing Transformer pooling outperforms applying MLP pooling (”how”).
>
> > “OCR can be slower for tasks where object-wise reasoning is not important”? It looks like all testbeds require some form of object-wise reasoning.
> >
>
> We apologize for the sentence misleading you. The sentence is to summarize our results shown in Figure 5 that SLATE showed slower or similar performance when compared to E2E CNN in terms of wall-clock time in Object Goal and Object Interaction tasks. Object Interaction task requires to learn the interaction between objects, but we thought it is different from the reasoning required for comparison tasks. We will change this sentence in our revised version.
>
> Additionally we agree that not every testbed must require some form of object-wise reasoning while we are targeting object-centric applications. It is one of the limitations in our study and modulating between these system 1 and 2 modes are interesting future research direction of the OCR community. We discussed it in further discussion section in Appendix D. Thank you for your comment!
>
> > “OCR provides disentangled representations for each object”? First, this property is never tested. Second, it is never compellingly shown that it is the reason for the good performance of OCR.
> >
>
> We apologize that this sentence could cause some confusion. What we want to say is that OCR provides separate representations for each object *between* slots, but we can see that this statement can be interpreted to mean OCR provides disentangled representations *within* each slot. This was not our intention and we will update the sentence in our revised version for clarity.
>
> For the claim that OCR provides separate representations for each object *******between******* slots, we refer to [1], where the slot representations are evaluated through robustness to changes affecting a single object in a scene where multiple objects exist. The authors find that changing one object in the scene does not affect the representation of other objects (Figure 7 in [1]).
>
> This disentangled representation can be the reason for good performance on the RL tasks because it is free from the binding problem in an environment with multiple objects [2]. This is analyzed in Question 1 of our paper.
>
> [1] Dittadi, Andrea, et al. "Generalization and robustness implications in object-centric learning." *arXiv preprint arXiv:2107.00637*
>  (2021).
>
> [2] Greff, Klaus, Sjoerd Van Steenkiste, and Jürgen Schmidhuber. "On the binding problem in artificial neural networks." *arXiv preprint arXiv:2012.05208*
>  (2020).
>
> > For those reasons, the main message I got from this paper strikes me as somewhat incremental: “object-centric representations work better for object-centric tasks”.
> >
>
> Please see the common response CR2 above. We would also like to add that it is not clear how non-OCR approaches work for the different object-centric tasks so it is important for the community to investigate this and establish some foundations.

---

> > ### Author Response · Authors · 2022-11-19
> > **Response to Reviewer EbVV (2/3)**
> >
> > > First, and as mentioned above, what about non object-centric tasks? How far do OCR lag behind non-OCR pretraining? This is important as it starts to provide some ground to answer the “when does OCR work” question.
> > >
> >
> > We agree it is good to test non object-centric tasks. As you mentioned, one can believe that OCR may lag behind non-OCR pretraining, but this has not been studied systematically and thoroughly. We would like to add though, that our analysis of the results of the Object Goal task compared with the results from the other tasks that require interaction or reasoning also provides insight into the “when does OCR work” question (Section 4.1.1).
> >
> > As an initial investigation into non object-centric tasks, we’ve designed a new “Non Object-Centric Goal task”, where the agent starts in a random position and simply must go to the center position of the scene. There could be some distractor objects in the scene, but they can be ignored by the agent. Thus, the model only needs to extract the agent position to solve the task. Since the representations of the distractors can be ignored and there is no target object to solve the task, we can say that this task is less object-centric than the other tasks. For GT, MLP pooling is used, similar to the Object-Interaction task.
> >
> > |  | Non Object-Centric Goal with no distractors | Non Object-Centric Goal with 4 distractors |
> > | --- | --- | --- |
> > | GT | 1.0 | 1.0 |
> > | E2E CNN | 1.0 | 0.998 $\pm$ 0.003 |
> > | SLATE | 1.0 | 1.0 |
> > | VAE | 1.0 | 0.208 $\pm$ 0.19 |
> > | CNN(VAE) | 0.407 $\pm$ 0.061 | 0.118 $\pm$ 0.045 |
> >
> > Without distractors, every model except CNN(VAE) can solve the task perfectly. The reason why CNN(VAE) is not working well must be investigated in future work but, one of our hypotheses is that it is hard to train Transformer pooling with a large number of patches through sparse reward. Note that we evaluate other CNN feature maps too for other tasks in response to Reviewer BK2Z.
> >
> > Even though there are distractors, GT, E2E CNN and SLATE can solve the task, while VAE fails to do this. This may be attributed to the binding problem [1]; similar results are shown in Table 2.
> >
> > From this result, even though the task requires less object-centric knowledge, SLATE can still solve it. However, this is one of the simplest non object-centric tasks, so it could be interesting to extend the difficulty of non object-centric tasks to where OCR fails.
> >
> > **Summary: Even for less object-centric task, OCR works comparably works with GT or E2E CNN.**
> >
> > [1] Greff, Klaus, Sjoerd Van Steenkiste, and Jürgen Schmidhuber. "On the binding problem in artificial neural networks." arXiv preprint arXiv:2012.05208
> >  (2020).
> >
> > > Second, are the E2E and VAE baselines trained on enough data and with enough variations? It would be help the paper if the authors could show that even with more data, VAE and E2E have already “plateaued” and so the benefits of OCR can’t be make up for. Similarly, what if E2E and VAE were trained on much larger datasets (eg: ImageNet) where learning OCR becomes much more challenging (because of the lack of object-centric labels).
> > >
> >
> > E2E CNN is not pretrained through dataset. VAE and OCR models are pretrained with same dataset consisting of 1 million frames. To evaluate that the dataset consists enough data with enough variations, we created larger 2D datasets including 2M and 3M frames, and validated the pretrained VAE with the larger dataset.
> >
> > |  | Object Goal | Object Int. | Object Comp. | Prop. Comp. |
> > | --- | --- | --- | --- | --- |
> > | VAE (1M dataset) | 0.674 $\pm$ 0.032 | 0.422 $\pm$ 0.04 | 0.684 $\pm$ 0.032 | 0.387 $\pm$ 0.035 |
> > | VAE (2M dataset) | 0.679 $\pm$ 0.032 | 0.4 $\pm$ 0.034 | 0.732 $\pm$ 0.036 | 0.389 $\pm$ 0.034 |
> > | VAE (3M dataset) | 0.672 $\pm$ 0.042 | 0.435 $\pm$ 0.045 | 0.691 $\pm$ 0.031 | 0.409 $\pm$ 0.039 |
> >
> > As we can see, the performances are similar, so we can say VAE is pretrained with enough variations already. We will update the paper with this result.
> >
> > For your next question about VAE pretraining with larger datasets (e.g., ImageNet), we do not believe that those datasets with natural images would be helpful to improve the RL performance in our synthetic environment. Our benchmark consists of synthetic circles, squares or triangles, which are not seen from the larger datasets (e.g., ImageNet), so while pretraining on those datasets can increase the span of the representations, they are out-of-distribution for our RL tasks. To increase the diversity of in-distribution data, we can create larger datasets from our benchmark, which is evaluated above.

---

> > > ### Author Response · Authors · 2022-11-19
> > > **Response to Reviewer EbVV (3/3)**
> > >
> > > > Third, and a more minor point, the benchmarking testbeds look a little bit simplistic (see Fig. 2). Do the insights in the paper carry to more realistic tasks such as real-world embodied AI (Habitat/AI2Thor)? The most realistic task in this paper (object-reach) is only ever used once (Fig. 6).
> > > >
> > >
> > > Please see the common response CR1 above.
> > >
> > > > The background section could include more details for at least one of the OCR method (eg: SLATE) to refresh the reader’s memory. It could also outline the main differences between all OCR methods.
> > > >
> > >
> > > We will add it in our revised version. Thank you for your comment.
> > >
> > > > Figures are unreadable for color-blind readers or when printed black-and-white.
> > > >
> > >
> > > Thank you for your feedback. We have updated the plots in the revised version.

---

### Official Review · Reviewer_BK2Z · 2022-10-25

**Confidence:** 4
**Correctness:** 3
**Technical Novelty And Significance:** 3
**Empirical Novelty And Significance:** 3
**Recommendation:** 6

**Clarity, Quality, Novelty And Reproducibility:**

* Clarity is very high. It’s really easy to follow and the choice of curves and results are great.
* Quality is also high, it is small in scope but everything seemed very well balanced and executed to me.
* Novelty is also good enough. The tasks aren’t dramatically different from existing ones, but they cover a good set, and the exact model comparison performed is clearly novel and will be valuable to refer to in the future.


**Strength And Weaknesses:**

1. The paper is clear and very well introduced. It makes its assumptions and targets clear, and the set of proposed tasks are clean and well chosen.
   1. The Spriteworld tasks cover a good extension of capabilities, and if object-centric models fail on this dataset I am not sure how one would expect them to target more complex environments.
   2. The Robotics task is a nice simple increase in complexity, but obviously one would prefer to have more of these (perhaps closer to the scope of MetaWorld [1]?), and more complex visuals.
2. I think the models selected are good, and I found the baselines to be appropriate, but I was expecting the E2E CNN to be used more.
   1. What is the performance of E2E CNN in Table 2b?
   2. I am not entirely sure that CNN(VAE) is that useful, it is not really something that people use?
   3. You could have used other types of “end of convolutional stack as objects” methods, like what’s used in SPACE, RelationNets, or even use the Conv backbone used by Slot Attention directly?
3. It might be important to discuss the effect of the training set used for the object-centric representations.
   1. In the current tasks, a random policy might be sufficient, but this is usually not the case otherwise.
   2. You could introduce more “on-policy” datasets (akin to Dreamer), and see how this changes results.
   3. This would also allow you to “count” or not the samples used for this in your sample efficiency assessment (e.g. consider starting from an empty dataset and training the representation online like Planet/Dreamer. It might fail, but if it works the numbers would be better?)
4. I really liked the presentation of Section 4. It is clear, hypothesis-driven and very well executed.
   1. Figure 3 is clean and easy to follow.
   2. Figure 4 somehow confused me, because I was trying to find the same columns as in Figure 3. Could you reorder/transpose it to match the structure of Figure 3 perhaps?
5. The text in Section 4.2 describing Table 2 does not seem to correspond to the content of Table 2?
   1. It feels like this is a leftover of a draft and should be updated. The table indicates better results than the main text.
6. I also really liked the observations in Appendix B.3, might be nice if this could be brought into the main text more strongly?

[1] https://meta-world.github.io/


**Summary Of The Paper:**

This paper proposes a set of simple RL tasks meant to assess the benefits of using object-centric representations for RL. They compare the performance of 3 recent object-centric representation learning methods (IODINE, Slot Attention and SLATE) as features for a simple PPO agent solving these tasks, assessing details of where they help or not.

**Summary Of The Review:**

I am quite torn about this paper, because even though I feel it is well executed and proposes something that would be quite useful to build upon and for the community to use, it is perhaps too limited in scope for ICLR (although there is past evidence that similar papers got accepted [1]). If that was NeurIPS, I would recommend submission to the Dataset and Benchmark track, where it would clearly shine and I would heartily support it.


However, despite these current limitations, as a practitioner I would leverage this benchmark for my own work and I feel it would be a valuable paper to refer to or integrate into existing open-source projects. Hence, I would currently recommend to borderline accept, even though I understand this might not be possible.

[1] https://openreview.net/forum?id=1W0z96MFEoH

---

> ### Author Response · Authors · 2022-11-19
> **Response to Reviewer BK2Z (1/2)**
>
> First of all, thank you very much for your constructive feedbacks, especially with regards to the CNN feature maps and the “on-policy” dataset.
>
> > but obviously one would prefer to have more of these (perhaps closer to the scope of MetaWorld [1]?), and more complex visuals.
> >
>
> Please see the common response CR1 above.
>
> > I think the models selected are good, and I found the baselines to be appropriate, but I was expecting the E2E CNN to be used more.
> >
> >
> > What is the performance of E2E CNN in Table 2b?
> >
>
> We originally didn’t compare E2E CNN for the tasks where there are fewer number of objects, because E2E CNN already showed good performance with more objects, but to respond to this question, we tested E2E CNN for these tasks. The results are in the below table.
>
> | task | #objs | E2E CNN | SLATE | VAE |
> | --- | --- | --- | --- | --- |
> | Object-Goal | 1 | $0.998 \pm 0.003$ | $0.997 \pm 0.01$ | $\mathbf{0.999} \pm 0.00$ |
> | Object-Goal | 3 | $0.985 \pm 0.01$ | $\mathbf{0.992} \pm 0.01$ | $0.686 \pm 0.02$ |
> | Object-Int | 1 | $0.984 \pm 0.008$ | $ \mathbf{0.99} \pm 0.01$ | $0.971 \pm 0.01$ |
> | Object-Int | 3 | $0.757 \pm 0.03$ | $\mathbf{0.859} \pm 0.03$ | $0.345 \pm 0.08$ |
>
> As we expected, for fewer number of objects (when the number of object is 1), E2E CNN works well.
>
> > I am not entirely sure that CNN(VAE) is that useful, it is not really something that people use?
> You could have used other types of “end of convolutional stack as objects” methods, like what’s used in SPACE, RelationNets, or even use the Conv backbone used by Slot Attention directly?
> >
>
> CNN(VAE) is not typically used as you pointed out, but we wanted to compare OCR pretraining with CNN feature map pretraining. To answer your question, we also evaluated the following representations:
>
> *CNN feature map from E2E CNN [1,2]*: Instead of using the distributed representation from the MLP on top of the CNN features, as is done with the E2E CNN model, we use the CNN feature map itself. This end-to-end learning of CNN feature map is used for Relational Networks [1] and Relational Reinforcement Learning [2]. For pooling, Transformer is used.
>
> *CNN feature map from SLATE [3]*: As another pretrained CNN feature map, we used the frozen CNN feature map from the pretrained SLATE encoder. Frozen CNN feature map from pretrained OCR is used in [3]. In the paper, the CNN feature map is encoded through another CNN which is trained through reward signal. We follows the architecture, but we evaluated also when applying Transformer on top of another CNN for pooling.
>
> *Multiple E2E CNNs [4, 5]*:  As previous works [4, 5] did, we evaluated the multiple CNN encoders that are trained through the reward signal. We used 5 encoders, and Transformer pooling is applied on top of the distributed representations from each encoder.
>
> The hyperparameters of Transformer pooling layer and policy are same to that of SLATE. The hyperparameters of MLP pooling layer are same to that of VAE.
>
> |  | Object Goal | Object Int. | Object Comp. | Property Comp. |
> | --- | --- | --- | --- | --- |
> | E2E CNN | 0.983 $\pm$ 0.015 | 0.899 $\pm$ 0.033 | 0.972 $\pm$ 0.007 | 0.208 $\pm$ 0.037 |
> | CNN Feat from E2E CNN | 0.973 $\pm$ 0.012 | 0.307 $\pm$ 0.415 | 0.22 $\pm$ 0.03 | 0.185 $\pm$ 0.038 |
> | Multiple E2E CNNs | 0.987 $\pm$ 0.005 | 0.222 $\pm$ 0.209 | 0.975 $\pm$ 0.016 | 0.599 $\pm$ 0.053 |
> | SLATE | 0.977 $\pm$ 0.006 | 0.963 $\pm$ 0.021 | 0.982 $\pm$ 0.011 | 0.978 $\pm$ 0.013 |
> | CNN Feat from SLATE with Transformer pooling | 0.972 $\pm$ 0.011 | 0.01 $\pm$ 0.0157 | 0.219 $\pm$ 0.037 | 0.218 $\pm$ 0.025 |
> | CNN Feat from SLATE with MLP pooling | 0.987 $\pm$ 0.008 | 0.647 $\pm$ 0.256 | 0.216 $\pm$ 0.029 | 0.224 $\pm$ 0.028 |
>
> CNN feature maps with Transformer pooling fail to solve any tasks except for the Object Goal task. This should be investigated more in future work, but one of our hypotheses is that it can be difficult to train the Transformer pooling with a large number of patch representations through the reward signal. One piece of evidence for this is that the model using MLP pooling on the CNN feature map from SLATE works better than the model with Transformer pooling on the same feature map for the Object Interaction task.
>
> Multiple E2E CNNs outperform E2E CNN except for the Object Interaction task. We hypothesize that this is because the encoder model size is much larger than E2E CNN and the reward from the Object Interaction task is much sparser than other tasks. Another interesting result is that Multiple E2E CNNs solve the Property Comparison task somewhat. From this, we can expect that through multiple encoders, the model can represent object-wise information as shown in [4, 5], but it is not perfect.
>
> **Summary: CNN feature maps are usually worse than the distributed representations on similar architecture because it is hard to train Transformer with a large number of patch representations. Multiple encoders can solve the comparison tasks somewhat, but not perfectly.**

---

> > ### Author Response · Authors · 2022-11-19
> > **Response to Reviewer BK2Z (2/2)**
> >
> > We updated our revised version with this results, thank you for your questions!
> >
> > [1] Santoro, Adam, et al. "A simple neural network module for relational reasoning." *Advances in neural information processing systems*
> >  30 (2017).
> >
> > [2] Zambaldi, Vinicius, et al. "Relational deep reinforcement learning." *arXiv preprint arXiv:1806.01830*
> >  (2018).
> >
> > [3] Heravi, Negin, et al. "Visuomotor Control in Multi-Object Scenes Using Object-Aware Representations." *arXiv preprint arXiv:2205.06333*
> >  (2022).
> >
> > [4] Kipf, Thomas, Elise Van der Pol, and Max Welling. "Contrastive learning of structured world models." *arXiv preprint arXiv:1911.12247*
> >  (2019).
> >
> > [5] Watters, Nicholas, et al. "Visual interaction networks: Learning a physics simulator from video." *Advances in neural information processing systems*
> >  30 (2017).
> >
> > > It might be important to discuss the effect of the training set used for the object-centric representations.
> >     1. In the current tasks, a random policy might be sufficient, but this is usually not the case otherwise.
> >     2. You could introduce more “on-policy” datasets (akin to Dreamer), and see how this changes results.
> >     3. This would also allow you to “count” or not the samples used for this in your sample efficiency assessment (e.g. consider starting from an empty dataset and training the representation online like Planet/Dreamer. It might fail, but if it works the numbers would be better?)
> > >
> >
> > To evaluate OCR pretraining, we limited the tasks in the scope where a random policy is sufficient to collect enough diverse observations. However, we agree that the “on-policy” dataset setting is an interesting topic, and we discussed it in the further discussion section in Appendix D.
> >
> > > Figure 4 somehow confused me, because I was trying to find the same columns as in Figure 3. Could you reorder/transpose it to match the structure of Figure 3 perhaps?
> > The text in Section 4.2 describing Table 2 does not seem to correspond to the content of Table 2?
> > >
> >
> > Thank you for your feedback. We updated them on our revised version. To match the structure, we added the unseen object test on Object-Interaction task.
> >
> > > I also really liked the observations in Appendix B.3, might be nice if this could be brought into the main text more strongly?
> > >
> >
> > Thank you for your comment. However, due to the space limitation, we leave it in Appendix. If we could make some space, we will bring it to main text.

---

### Official Review · Reviewer_YJeG · 2022-10-26

**Confidence:** 4
**Correctness:** 3
**Technical Novelty And Significance:** 2
**Empirical Novelty And Significance:** 2
**Recommendation:** 5

**Clarity, Quality, Novelty And Reproducibility:**

Clarity

The organization of the paper is logical and facilitates understanding. I would have liked for a bit more detail of the models to be present in the main text, e.g. the precise form of the OCR representations before pooling (presumably a sequence of slots/tokens).

Quality

The execution of the work is largely good. The significance is somewhat limited, as argued above.

Novelty

The main novelty of this work lies in the formulation of the study's research questions and the creation of the Spriteworld and CausalWorld tasks. There is no technical novelty otherwise.

Reproducibility

The methods are described in a reasonable level of detail. Source code is not provided or promised, however.

**Strength And Weaknesses:**

Strengths

This paper sets out to empirically validate important motivating assumptions underlying the field of object-centric representation learning. The methods are clearly presented and the interpretation of the results is largely sensible.

Weaknesses

The toy nature of the experiments restricts the significance of the study to environments with similarly limited complexity. That is, part of the answer to the "When" in the title must include "when OCRs are essentially perfect". At the very least, the claims made in the paper should be so qualified. And unfortunately, I'm not sure that there existed much doubt about the efficacy of OCRs in this setting.

The descriptions of the proposed tasks include a large number of plausible-sounding yet unsupported statements about necessary conditions for solving each task. For example, for Object Comparison, the agent does not necessarily need to compare all pairs of objects; it could succeed by simply learning to represent how much different colors appear in the image, and move towards the rarest color. Unfortunately, the inaccuracy in these statements propagates to the interpretation of the results, resulting in some overclaiming.

**Summary Of The Paper:**

This paper investigates multiple hypotheses regarding the use of object-centric representations (OCRs) for reinforcement learning. Specifically, they consider object-centric representations that are pre-trained using unlabeled interaction data in the environment before being frozen for use in RL. The hypotheses are that OCR pretraining improves RL sample efficiency (H1), that OCR pretraining improves performance on tasks involving relational learning (H2), and that OCR pretraining helps facilitate out-of-distribution generalization to new objects or combinations of objects (H3). To assess these hypotheses, the authors construct 4 2D interactive navigation tasks in Spriteworld and 1 robotic manipulation task in CausalWorld. The authors deliberately limit the complexity of the involved objects to assess OCR in a best-case setting. The authors consider the SLATE, IODINE, and Slot-Attention OCR encoders, and $\beta$-VAE (using the VAE latent or CNN feature map), and end-to-end learned CNN as non-structured baselines. Finally, ground truth low-dimensional state is also compared to. The authors present results that largely support the 3 hypotheses, and also contribute results tackling 5 other research questions.

**Summary Of The Review:**

Overall, this paper is well-written and well-executed, but as an empirical study of prior methods its significance is severely limited by its chosen scope. I currently recommend borderline rejection, but with low confidence since I could be swayed by more positive assessments of impact from other members of the reviewing team with more skin in the field of object-centric representation learning.

---

> ### Comment · Reviewer_YJeG · 2022-11-18
> **Review update**
>
> Given that my fellow reviewers seem to agree with my major criticisms and that the authors have not responded, I have increased my confidence in my main review.

---

> > ### Author Response · Authors · 2022-11-19
> > **Response to Reviewer YJeG**
> >
> > Thank you very much for your valuable comments and sorry for late response. We will share our response to your questions and discuss your concerns.
> >
> > > The toy nature of the experiments restricts the significance of the study to environments with similarly limited complexity. That is, part of the answer to the "When" in the title must include "when OCRs are essentially perfect".
> > >
> >
> > Please see the common response CR1 above.
> >
> > > At the very least, the claims made in the paper should be so qualified. And unfortunately, I'm not sure that there existed much doubt about the efficacy of OCRs in this setting.
> > >
> >
> > Please see the common response CR2 above.
> >
> > > The descriptions of the proposed tasks include a large number of plausible-sounding yet unsupported statements about necessary conditions for solving each task. For example, for Object Comparison, the agent does not necessarily need to compare all pairs of objects; it could succeed by simply learning to represent how much different colors appear in the image, and move towards the rarest color. Unfortunately, the inaccuracy in these statements propagates to the interpretation of the results, resulting in some overclaiming.
> > >
> >
> > We agree that we overclaimed the necessary conditions for Object Comparison task. Even though some reasoning is required to solve the task, it does not have to necessarily be object-wise. To navigate to the target object, object-wise knowledge is required but it is not meant that object-wise reasoning is required.
> >
> > To cover our claim, we updated the Object Comparison task to navigate to the unique color or shape. In the environment, there will be a single object with unique color or shape, which is the target object to be collected by the agent. Different from Property comparison, other 3 objects are same in this task to evaluate the object-wise reasoning.
> >
> > |  | Previous Object Comparison Task | New Object Comparison Task |
> > | --- | --- | --- |
> > | GT | 0.976 $\pm$ 0.015 | 0.967 $\pm$ 0.015 |
> > | E2E CNN | 0.977 $\pm$ 0.015 | 0.421 $\pm$ 0.159 |
> > | SLATE | 0.973 $\pm$ 0.02 | 0.984 $\pm$ 0.01 |
> > | VAE | 0.703 $\pm$ 0.035 | 0.381 $\pm$ 0.06 |
> > | CNN(VAE) | 0.22 $\pm$ 0.021 | 0.295 $\pm$ 0.071 |
> >
> > GT performance is a little degraded, and SLATE performance is improved a little. Although, E2E CNN and VAE performances largely deteriorate. As you commented, perhaps previous task just requires to find the rarest color, so the agents with distributed representation could solve it, but they fail on new task where shape (object-wise information) is required to be solved.
> >
> > Thank you for pointing out this issue. We will update it.
> >
> > **Summary: Previous object comparison task was overclaimed. We redesign the task to cover our claim, and only OCR shows comparable performance with GT performance.**
> >
> > > I would have liked for a bit more detail of the models to be present in the main text, e.g. the precise form of the OCR representations before pooling (presumably a sequence of slots/tokens).
> > >
> >
> > We will add it in our revised version. Thank you for your comment.
> >
> > > Source code is not provided or promised, however.
> > >
> >
> > We will release our codes with our benchmark as mentioned at the last of Introduction section.

---

### Official Review · Reviewer_JnXb · 2022-10-27

**Confidence:** 2
**Clarity, Quality, Novelty And Reproducibility:** The paper is very clear. Most impleme…
**Correctness:** 3
**Technical Novelty And Significance:** 2
**Empirical Novelty And Significance:** 3
**Recommendation:** 5

**Strength And Weaknesses:**

The paper presents a nice analysis on the environments and tasks it studies. It shows that OCR outperforms end-to-end distributed representations on relational tasks and scenes with many objects but not on simple object goal tasks, which aligns with our expectation of OCR.

However, my main concern is that the experiments only concern two simple synthetic environments. I am not sure if this can be considered as a comprehensive study on whether OCR pre-training is effective for reinforcement learning.

The experiments are als similar to the experiments in COBRA (Watters et al.), which also uses Spriteworld. It would be good if the authors could highlight the difference.

Watters et al. COBRA: Data-Efficient Model-Based RL through Unsupervised Object Discovery and Curiosity-Driven Exploration

---

Minor question:

Why would OCR models outperform the GT model in second figure in Figure 3? I would image GT as an upper-bound.

How is the GT state embedded? What does the ground truth state include?

**Summary Of The Paper:**

The paper provides an empirical evaluation of whether object-centric representation pre-training is useful for RL learning. They find that OCR pre-training generally delivers better and more data-efficient model, also allowing generalization to unseen settings (e.g., an unseen number of objects).


**Summary Of The Review:**

Overall, while I think the authors provide nice analysis on the studied environment. However, I am not sure if the experiments can be considered as comprehensive enough to support some of the rather general claims the authors made.

---

> ### Author Response · Authors · 2022-11-19
> **Response to Reviewer JnXb**
>
> Thank you very much for your careful review. Your comments on details, such as emphasizing the difference from the COBRA experiments, are very helpful in making our paper more concrete.
>
> > However, my main concern is that the experiments only concern two simple synthetic environments. I am not sure if this can be considered as a comprehensive study on whether OCR pre-training is effective for reinforcement learning.
> >
>
> Please see the common response CR1 above.
>
> > The experiments are also similar to the experiments in COBRA (Watters et al.), which also uses Spriteworld. It would be good if the authors could highlight the difference.
> >
>
> The COBRA paper also handled OCR for RL with Spriteworld environments. However, there are several differences from ours. For modeling, the paper uses a simple search rather than policy learning. In contrast, our study studies a model-free RL setting using PPO. While distractors exist in their task, they can be ignored. However, in our tasks, the agent must move to or interact with the target object while avoiding the distractors. This requires extracting not just the target object information from the scenes but also the distractor object information. In addition, while they evaluated some comparison tasks through the clustering task, they didn’t investigate property-level comparison as we did. We will add a discussion of these differences in the Introduction section. Thank you for pointing this out.
>
> > Why would OCR models outperform the GT model in second figure in Figure 3? I would image GT as an upper-bound.
> >
>
> Good catch. This result is when the PPO entropy coefficient was set to 0.1. This hyperparameter makes the action noisier, which can lower the success rate when the object is near a wall. After some tuning, we found better results by lowering the entropy coefficient to 0 like used for other methods, which we’ve updated in the revised version of the paper. With this change, the ground truth learns more quickly than the other models, as we might expect. However, even with this change, GT is still not an upper-bound of other models yet in terms of final success rate after 2 million steps. When examining the trajectories of the policy, the agent trained from GT often failed while trying to pass between objects without considering the size of the objects when several objects were close to each other. The agent trained from SLATE representations, on the other hand, correctly navigates around the objects, leaving enough room between the agent and the objects. This is probably because size information is given as an index in GT, so it was difficult to use with distance information. However, SLATE doesn’t seem to have this problem because it gets the object representation from the scene.
>
> > How is the GT state embedded? What does the ground truth state include?
> >
>
> The GT states are embedded in the form of $[\text{number of objects} \times \text{state per object}]$.
>
> For 2D tasks, the state per object consists of 5 dimensions for color index, shape index, size index, and position (x and y coordinates).
>
> For the 3D task, the state consists of 37 dimensions per object. The robot state is 27 dimensions (9 dimensions for joint positions, 9 dimensions for joint velocities, and 9 dimensions for end effector positions). Each object is 9 dimensions (3 for cartesian position, 3 for size, and 3 for color (RGB)). To represent the ground truth state of both robot state and objects in terms of object slots, these 36 dimensions are concatenated together with an additional dimension indicating whether the slot is the robot arm or an object.
>
> We added this description in Appendix E.1.1 of the paper.

---

### Author Response · Authors · 2022-11-19
**Common Response**

First of all, thank you very much for your sincere responses. Before sharing our responses to the questions of each reviewer, we will discuss a few shared concerns here first.

- (CR1) Synthetic nature of the tasks:

    All reviewers commented on the synthetic nature of the tasks. Reviewer JnXb commented that these simple synthetic environments couldn’t be enough to study whether OCR pretraining is good for RL comprehensively and Reviewer YJeG commented that this simplicity limited the significance of this study. Reviewer BK2Z commented that more complex visuals would be preferable. Reviewer EbVV commented on investigating more realistic tasks such as real-world embodied AI.

    Our benchmark can seem very synthetic, but it is because current OCR models still struggle with more visually complex scenes. Thus, we chose visually simple scenes to ensure the downstream reinforcement learning performance is not affected by poor segmentation quality. This allows us to probe more specific aspects of the reinforcement learning task to assess where pretraining OCR is most beneficial. In order to investigate the case where segmentation quality is not perfect, we also ran experiments on the robotics Object Reaching Task, which we discuss in Q3. We agree that investigating OCR on more complex and realistic environments is a promising direction for future work, especially as unsupervised OCR models continue to improve.

- (CR2) Signficance of the work:

    Several reviewers commented on the significance of the studies. Reviewer JnXb stated that they were not sure that there existed much doubt about the efficacy of OCRs in our experiment settings. Reviewer EbVV commented that the work seems somewhat incremental.

    We agree that there is a general belief that OCR should be beneficial to downstream RL tasks, so the results of some of our investigations may not be very surprising. In fact, this is exactly our motivation for carrying out this study. Many previous works [1, 2, 3, 4] mentioned RL as a downstream task for OCR, but there is actually no clear evidence supported by systematic analysis for how OCR will perform in RL. By running our experiments, we provide empirical evidence for several assumptions such as how OCR affects the generalization of the agent, which tasks OCR are better for than single vector representations, and which OCR model is better than others on RL tasks. Furthermore, our study uncovers several results that may not be as intuitive, such as OCR being worse than an end-to-end learned CNN policy in terms of wall-clock time for the Object-Interaction task.

    [1] Greff, Klaus, et al. "Multi-object representation learning with iterative variational inference." *International Conference on Machine Learning*
    . PMLR, 2019.

    [2] Burgess, Christopher P., et al. "Monet: Unsupervised scene decomposition and representation." *arXiv preprint arXiv:1901.11390*
     (2019).

    [3] Engelcke, Martin, et al. "Genesis: Generative scene inference and sampling with object-centric latent representations." *arXiv preprint arXiv:1907.13052*
     (2019).

    [4] Lin, Zhixuan, et al. "Improving generative imagination in object-centric world models." *International Conference on Machine Learning*
    . PMLR, 2020.

---

### Author Response · Authors · 2022-11-19
**List of updates in our revision**

- highlight the difference from COBRA paper experiment in Introduction section (commented by JnXb).
- In Figure 3, GT performance is updated.
- In Figure 4, for Object Interaction task, unseen object test is added.
- In Table 1 and Table 3, unseen combination test is updated with the results from more diverse results.
- In Figure 7, OCR models comparison for Object Interaction task is updated.
- “on-policy” dataset setting is discussed as the future direction in Appendix D (commented by BK2Z).
- In Appendix D.1.1, GT descriptions are added (commented by JnXb).
- In Appendix C.3, we updated additional CNN feature map results (commented by BK2Z).
- Updating conclusion as commented by EbVV.
    - Update the sentence “it can be slower for tasks where object-wise reasoning is not required”.
    - Update this sentence “OCR provides disentangled representations for each object”.
- Investigating the tasks requiring non object-centric reasoning is discussed in Appendix D (commented by EbVV).
- Plot styles are changed (commented by EbVV).

---

### Decision · Program_Chairs · 2023-01-20

**Decision:**

Reject

**Justification For Why Not Higher Score:**

The authors have added some clarifications and some additional results for small variations of their existing setup during the rebuttal, but the main concern around the limited scope of the investigation that the authors chose for this paper remains.

**Justification For Why Not Lower Score:**

N/A

**Metareview: Summary, Strengths And Weaknesses:**

This paper investigates the utility of object-centric representation (OCR) pretraining for policy learning in multi-object environments using a reinforcement learning (RL) objective. The paper systematically compares OCR pretraining against non-OCR baselines (e.g. CNNs) on a set of synthetic benchmark tasks: multiple tasks using SpriteWorld, specifically designed for this study, and one task on a synthetic robotic reacher task.

All reviewers agree that this paper is well-written, generally of high quality, and addresses an interesting problem. The reviewers positively highlight the hypothesis-driven experimental investigation in the paper. The paper is generally novel, in that the exact setup chosen in the paper has not been addressed in prior work, but has limited novelty otherwise: related systematic investigations for policy learning exist: prior work [1] has already investigated object-centric representations for control (directly compared to autoencoder and contrastive models) for a related robotics task with very similar conclusions. Although their study uses an implicit behavioral cloning objective instead of a reinforcement learning (RL) objective to learn a policy, this reduces the significance of performing a similar study using RL.

The main (and major) concern shared by all reviewers is that the paper has chosen a very limited scope with potentially low significance for the community. The title of the paper suggests a broader scope than what the paper covers: the environments are of lower complexity than those covered in typical prior work on visual pre-training for control (see e.g. [1,2]): the four SpriteWorld tasks — while being a good test-bed for the hypothesis stated in the paper — consist solely of uni-colored sprites on black background and have unclear relevance for real-world problems. The proposed Object-Reach. task is a good step-up in visual complexity, but is otherwise still very simple with uni-colored cube objects, with the target object being always the same (blue cube). It is unclear how insights gained from these environments would translate beyond these setups which are currently of limited significance. As highlighted by the reviewers, the paper otherwise has limited technical novelty as it primarily consists of an empirical comparison of existing methods on newly designed benchmark tasks.

While the authors have added some clarifications and some additional results for small variations of their existing setup, this main concern remains. I think the direction this paper is taking is interesting and should be further expanded upon: the paper in its current form would heavily benefit from an expanded scope before being considered for publication, but is otherwise of high quality.

To improve the scope/significance of the paper, the authors could for example explore the following directions:
* The authors could consider adding more diverse non-OCR baselines, such as the one considered in [1] or a MAE baseline (e.g. the model from [2])
* Investigate OCR pre-trained representations for tasks with non-obvious object structure in the visual observation
* Investigate tasks where models like SLATE would currently still struggle to understand the limitations and headroom for current approaches


[1] Heravi et al., “Visuomotor Control in Multi-Object Scenes Using Object-Aware Representations” (2022)
[2] Xiao et al., “Masked Visual Pre-training for Motor Control” (2022)